# Parenchymal Sparing Surgery for Lung Cancer: Focus on Pulmonary Artery Reconstruction

**DOI:** 10.3390/cancers14194782

**Published:** 2022-09-30

**Authors:** Cecilia Menna, Erino Angelo Rendina, Antonio D’Andrilli

**Affiliations:** Division of Thoracic Surgery, Sant’Andrea Hospital, Faculty of Medicine and Psychology, Sapienza University of Rome, 00189 Rome, Italy

**Keywords:** pulmonary artery reconstruction, lung cancer, sleeve resection

## Abstract

**Simple Summary:**

Reconstruction of the pulmonary artery associated with lobectomy for the radical resection of lung cancer is a safe and effective therapeutic option that may allow radical resection when lobectomy is not technically feasible, avoiding pneumonectomy. This review addresses some controversial aspects concerning the intraoperative and perioperative management of a sleeve resection with pulmonary artery reconstruction that may influence the outcome. Pulmonary artery reconstruction associated with lobectomy is a safe and viable parenchymal sparing intervention to radically treat lung cancer, allowing better long-term survival, lower perioperative morbidity and mortality rates and functional benefits if compared with PN.

**Abstract:**

Reconstruction of the pulmonary artery (PA) associated with lobectomy for the radical resection of lung cancer has been progressively gaining diffusion in lung cancer surgery as a safe and effective therapeutic option that may allow radical resection when lobectomy is not technically feasible, avoiding pneumonectomy. There are some controversial aspects concerning the intraoperative and perioperative management of a sleeve resection with PA reconstruction that may influence the outcome. In the present article, the authors have analyzed some of the main technical and oncological aspects to take stock of what they have learned from their lung-sparing operations experience over time. PA reconstruction may require prosthetic materials including different options with variable cost. A main concern in vascular reconstructive procedures is avoiding tension on the anastomosis. When PA reconstruction is required, appropriate anticoagulation management is crucial. Results from the main literature data confirm the reliability of lobectomy associated with PA reconstruction in terms of perioperative morbidity and long-term survival. Sleeve lobectomy and PA reconstruction can be performed safely and effectively even after induction therapy.

## 1. Introduction

Lobectomy combined with resection and reconstruction of the bronchus, pulmonary artery (PA), or both has been shown to be a viable option for the treatment of lung cancer in the past three decades [1,2]. In most cases, this intervention is defined as Sleeve Lobectomy (SL), since a sleeve resection of the bronchus and/or the PA is required. SL for lung cancer is indicated when the origin of a lobar bronchus and/or the origin of the lobar branches of the PA are infiltrated by the tumor (Figure 1) but not so far to require a pneumonectomy (PN). In addition, SL may be indicated when the bronchus and/or the PA is involved by N1 lymph node infiltration, which is more common in case of a left upper lobe tumor which often requires a combined bronchial and vascular reconstruction. After induction therapy, even in the absence of neoplastic infiltration, when non-removable fibrotic tissue involves the PA and/or the bronchus, reconstructive procedures may also be indicated [3]. The present article aims to focus on reconstructive procedures of the PA associated with lobectomy after vascular en bloc resection.

The first case of tangential resection and direct suture repair of the PA for lung cancer was reported by Allison in 1952 [4]. Later, further descriptions of PA reconstructive procedures were made by Thomas, Parelman, and Wurning between 1956 and 1967 [5]. Gundersen in 1967 reported for the first time the successful results of sleeve resection and anastomotic reconstruction of the PA in two patients with left upper lobe tumor [6]. In 1971, Pichlmeier and Spelberg described four cases of combined bronchial and vascular sleeve resections without complications [6]. The series of 37 PA sleeve resections published by Vogt-Moykopf in 1986 proved that this intervention was feasible with an acceptable morbidity rate and good long-term prognosis [7]. However, since the first PA reconstructions for lung cancer resection were reported in the early 1950s, significant technical advances and increasing experience over time have progressively allowed the achievement of excellent clinical and oncologic results, resulting in wide consensus on the use of parenchymal-sparing procedures.

These surgical procedures may be necessary to avoid PN in patients with compromised heart and/or pulmonary function. However, the benefits of parenchymal preservation have been proved even in patients without cardiopulmonary impairment by recent studies [8,9,10].

Lobectomy associated with bronchial and vascular reconstruction presents higher technical difficulty than standard major lung resection, but recent experience suggests that these reconstructive interventions are associated with better overall survival, postoperative morbidity and mortality compared to PN [11,12,13,14,15,16].

## 2. Indications and Preoperative Evaluation

The indication for an SL and PA reconstruction for lung cancer is well defined: a primary tumor or N1 nodes infiltrating the origin of the lobar branches of the vessel but not as far as to make a PN unavoidable. Establishing the definitive indication for a reconstructive procedure of the PA may be not always possible preoperatively. The most appropriate diagnostic exam to identify the vascular involvement is the Computed Tomography (CT) scan with contrast medium. Angiography and Magnetic Resonance Imaging could add further details in order to better define the infiltration pattern. Nevertheless, the extension of the PA infiltration is not always clearly defined at preoperative imaging, and the definitive decision is made intraoperatively. Not infrequently, incongruities between radiological evidence and intraoperative findings, due to over- or underestimation of the vascular involvement by the CT scan, may lead to an incorrect indication.

CT scan is also a valid diagnostic method to evaluate the nodal status of the disease. If enlarged mediastinal lymph nodes are found, histologic or cytologic assessment by mediastinoscopy or Trans-Bronchial Needle Aspiration (TBNA) is indicated. In case of N2 disease confirmation, induction therapy is indicated.

When a bronchial or/and vascular infiltration is suspected, bronchoscopy is a mandatory preoperative examination to be performed by one of the operating surgeons in order to anticipate the intraoperative appearance of the airway. An eventual presence of bronchial wall compression may suggest peribronchial tumor infiltration. This is crucial information, especially for the anatomical areas where the PA is adjacent and potentially involved.

If a bronchial reconstruction is performed, it is advisable to perform repeated bronchoscopies at the end of surgery, before discharge from the hospital, and during the follow-up period (usually after 1, 6 and 12 months).

The right and the left PAs can be involved to various longitudinal extents. This condition is crucial to assess the feasibility of a reconstructive procedure. Although contrast-enhanced CT scan is an effective tool to assess preoperatively the PA infiltration, this examination is generally not sufficient to correctly plan the vascular reconstructive procedure and to choose the appropriate technical option.

A tangential resection with direct suture could be appropriate to achieve a radical resection when a very limited infiltration of the arterial wall is found at surgical exploration. The reconstruction can be performed using a patch (of biological or synthetic material), avoiding a circumferential resection for larger defects ensuing from a greater vascular involvement, up to 30% or 40% of the vessel circumference. A sleeve resection and reconstruction by an end-to-end anastomosis, or by the interposition of a prosthetic conduit is indicated for more extended infiltration.

Thus, the reconstructive technique and the eventual choice of prosthetic material have to be decided intraoperatively in most cases.

Postoperative quality of life has been considered one of the most important factors when making the decision to perform an SL rather than a PN. In a number of studies [16,17], lung parenchyma sparing resections have been demonstrated to improve postoperative quality of life as the result of a superior pulmonary reserve, with a lower postoperative loss of FEV1 after SL showing statistical significance [18,19]. Adequate pulmonary and cardiac functional evaluation is mandatory in the preoperative setting.

Nonetheless, the decision to perform PN or SL is primarily made in order to obtain a complete (R0) resection, so considering both the oncological and physiological aspects. A large amount of clinical data prove that PN, particularly right PN, can be considered as a “disease itself”, with severe postoperative impairment of lung capacity, cardiac function, and quality of life. Thus, this intervention should be performed only when necessary to achieve a complete oncological resection.

## 3. PA Reconstruction

### 3.1. Technical Issues

The first phase of the surgical procedure consists of accomplishing the complete dominion of the proximal portion of the PA. Even though division of the superior pulmonary vein can simplify the exposure of the PA, transection of this vessel should be deferred until becoming aware of the feasibility of the procedure. The resection step begins once the main and distal PA, the bronchus, and both pulmonary veins have been properly prepared. However, to obtain a complete tumor resection, PN can be unavoidable when extended infiltration of the PA is present, as in the condition of a left upper lobe tumor infiltrating the concave surface of the PA from its origin down to the anterior–basal artery or, on the right side, in the case of posterolateral infiltration at the level of the artery for the superior segment of the lower lobe.

The adequateness of a limited tangential resection with a consequent direct suture for a minimal infiltration of the arterial wall has been reported by some authors [20].

Lobectomy with “en bloc” PA resection and reconstruction by an end-to-end anastomosis, a patch, or prosthetic conduit is required for more extended neoplastic involvement of the vessel. Additionally, residual tumor or scarring tissue ensuing after induction therapy and involving the PA may require a sleeve resection.

When performing a sleeve resection with anastomotic reconstruction, it is advisable to standardize the intraoperative strategy. Accomplishing full control of the PA is a key intent. The resection phase should start when the PA, bronchus, and pulmonary veins are properly isolated. Firstly, the superior pulmonary vein is divided. Generally, the anatomical distance of the pulmonary veins from the tumor makes their control less challenging. Nevertheless, on the left side, the tumor may involve the upper lobe bronchus and the anterior portion of the fissure. An intrapericaldial control of the superior pulmonary vein can be easily achieved. For the reconstructive phase, the dissection in the interlobar fissure is crucial, which is mainly on the left side. Clamping of the PA is then performed. A dose of 1500 Units of heparin sodium (instead of the dose between of 3000 and 5000 Units used in the past) is administered intravenously [21]. Afterwards, the main PA is clamped proximally first and then distally with respect to the infiltrated vascular segment. “En bloc” vascular resection is then performed.

Patch reconstruction, end-to-end anastomosis or conduit interposition can be considered depending on the PA defect extent during the reconstructive phase. For all of the above-mentioned PA reconstruction procedures, running 5-0 monofilament nonabsorbable sutures are used [22,23,24]. After completing the reconstructive suture, the vascular clamp is partially opened before the suture is tied, so restoring the backflow and allowing air drainage. The suture is then tied, and the arterial clamp removed. Checking the suture line and prudently assessing lung re-expansion to exclude the occurrence of PA kinking or folding is mandatory before closing the chest. At the end of the reconstruction, especially if a bronchial sleeve is associated, it is useful to protect the suture line, placing a viable vital tissue between the artery and the bronchus. The intercostal muscle flap is the authors’ first choice [25]. Subcutaneous low-molecular-weight heparin administration is advisable for 7 up to 10 days after surgery.

### 3.2. Partial Resection and Patch Reconstruction

Patch reconstruction is a very versatile technique and can be used in a variety of situations, ranging from moderate infiltration involving the origin of segmental arteries to significant defects extended longitudinally on one face of the PA. The opposite side of the circumference of the PA has to be free from tumor; otherwise, a sleeve resection with end-to-end anastomosis or conduit interposition is required. In order to reduce the arterial clamping time, vascular reconstruction is performed before the bronchial anastomosis, when necessary. After the partial resection, an oval defect oriented along the PA axis ensues, even if the resected portion is round. This is caused by the tension that the lower lobe produces on the vessel. It is advisable to tailor the patch on the resected vascular portion rather than on the PA defect (Figure 2). Although several tissues are available for a patch repair, biologic materials, such as azygos vein, autologous pericardial tissue, and heterologous (bovine or porcine) pericardial tissue are preferable thanks to a higher biocompatibility. The harvesting of the autologous pericardial flap is performed anteriorly to the phrenic nerve, and the pericardial defect is left open (Figure 3). The patch is properly trimmed and then fixed to the margins of the artery wall defect by 2 stay sutures. In this phase, a certain grade of tension is necessary to find a potential excessive length of the patch; the tension would withdraw after unclamping. Since the edge of the pericardial tissue has the tendency to shrink and curl, the suture has to be performed meticulously; otherwise, the anastomosis could become far apart. The use of bovine pericardial tissue can avoid some of the technical difficulties of harvesting, trimming, and suturing the autologous pericardial tissue. The bovine pericardial tissue has less elasticity, with smooth and more rigid edges. The advantages of bovine pericardial tissue are: no need for an additional harvesting procedure and the unlimited quantity of the tissue. The porcine pericardium has similar technical characteristics but reduced thickness compared to the bovine one. A 5-0 or 6-0 monofilament nonabsorbable running suture is used, starting from the top and proceeding to the bottom on the right side, while grasping and stretching the patch, ongoing from the bottom to top on the left side. The inferior stay suture is not tied, since it is used exclusively to hold the patch in place and is then removed when the suture line arrives at the level.

To improve technical features of the autologous pericardium, the authors have devised and described in the past an intraoperative method of fixation of the patch by a glutaraldehyde-buffered solution [26]. The glutaraldehyde preservation of the pericardium minimizes its tendency to retract and curl, thus allowing an easier vascular reconstruction, and reducing the risk of bleeding from the patch suture related to the elastic recoil of the autologous pericardium.

### 3.3. Sleeve Resection and Reconstruction by End-to-End Anastomosis

When 50% or more of the PA circumference is infiltrated by the tumor, a sleeve resection is always mandatory. After the resection, frozen section histology on vascular margins should be always performed, and if tumor infiltration is still present, PN is required.

When a bronchial anastomosis is associated, PA reconstruction is usually performed after its completion to minimize the manipulation of the vessel. Moreover, the exposure of the bronchial stumps is better when the artery is divided. Finally, after completion of the bronchial anastomosis, the tension on the vascular anastomosis is reduced because the bronchial axis is shortened. Technical issues regarding bronchial reconstruction after SL have been extensively reported in previous publications by the present authors [6,20,21,22] and are not repeated in this article due to its specific focus on PA reconstruction.

Regular and smooth margins (both proximally and distally) are desirable once dividing the artery, even at the expense of some loss of tissue. This can facilitate the proper placement of the stitches and adequate vascular stumps approximation. Furthermore, regular suture margins make the correction of the eventual caliber discrepancy easier. Due to the elasticity of the arterial wall, caliber discrepancy never represents a technical problem. By uplifting the lower lobe while suturing, the gap between the two arterial ends can be reduced. The posterior portion of the anastomosis is performed as a first reconstructive step. Afterwards, the anterior part of the suture (which is the simpler phase) is accomplished. Once the anastomosis has been completed, the suture is tied after removal of the distal clamp to restore backflow and to allow air drainage. Once the anastomotic suture is completed and before it is tied, the distal vascular clamp is partially opened, thus restoring the backflow and allowing air drainage within the untied portion of the suture line. The suture is then tied, and the arterial clamp is removed. Any residual tension is released by the restoration of blood flow and the removal of the proximal clamp. A prosthetic conduit implant is indicated when the gap between the arterial stumps is considered excessive. A 5-0 or 6-0 monofilament nonabsorbable material is used to perform the anastomotic running suture, placing stitches meticulously to avoid potential future stenosis. The end-to-end anastomosis can be technically more demanding in case of higher tension between the vascular stumps and caliber discrepancy (Figure 4).

It is important to test the arterial axis and suture line after re-expansion of the residual lobe/lobes. Re-inflation of the lower lobe elevates the hilum, potentially determining a rotation and a kinking of the PA. The consequent distortion of the suture line may cause re-opening of bleeding sites that are not visible when the artery is stretched downward by the atelectatic lower lobe.

### 3.4. Sleeve Resection and Reconstruction by a Prosthetic Conduit

Conduit interposition may be necessary, after a sleeve resection of the PA, when an excessive distance between the two vascular stumps ensues. This condition would produce a high tension on an end-to-end anastomosis. Such a technical situation may occur more frequently on the left side, when a left upper lobe tumor infiltrates the PA extensively but the lobar bronchus is not involved. This infrequent situation (PA sleeve without bronchial sleeve) may determine a long bronchial segment separating the two widely spaced PA stumps; thus, an end-to-end anastomosis can be not feasible. A tubular prosthesis generally of biological material, such as autologous or heterologous pericardial tissue [27,28,29] or pulmonary vein portion [30], can be used. Alternatively, other published reports have described the anecdotal use of cryopreserved allograft (PA or descending aorta) [31] harvested from multiorgan donors or saphenous vein autograft to reconstruct the PA [32]. Thanks to a better biocompatibility and a lower risk of thrombosis, biological materials are usually preferred. Autologous pericardial tissue or pulmonary vein conduits are fresh, cost-free, and biocompatible. Conversely, heterologous bovine or porcine pericardial tissues have inferior cost-effectiveness and biocompatibility, but they are extremely easy to use. During surgery, the autologous or heterologous pericardial membrane is cut to a rectangular shape and wrapped around a chest tube or a syringe of a proper caliber and sutured longitudinally. This suture was performed manually until some years ago and mechanically most recently. The same technique for the intraoperative creation of the conduit has been largely used by the authors for reconstruction of the Superior Vena Cava in oncologic operations [33,34,35]. When the autologous pericardium is used, the epicardial surface is oriented inside the conduit lumen.

A very interesting alternative for conduit reconstruction is the autologous pulmonary vein conduit when available (tumors not infiltrating the extrapericardial pulmonary vein portion), because this is an ideal substitute for the PA: it has adequate thickness and structural similarity with the arterial wall. Moreover, the elasticity of the two tissues is comparable.

A better proximal control of the superior pulmonary vein can be obtained by opening the pericardium when performing an upper lobectomy. Then, the superior vein is fully dissected, sutured proximally (at the atrial junction) with a vascular mechanical stapler and ligated distally at the level of its extralobar branches, thus dividing it and creating an autologous vein conduit of approximately 1.5 to 2.5 cm, which is preserved in saline solution for few minutes (Figure 5). Before going on with the reconstructive procedure, a frozen-section examination of the distal margin on the vein resection specimen is performed. Since the grade of elasticity of the venous tissue is comparable to that of the PA, it is prudent to adapt the length of the conduit based on the resected arterial segment. A running 5-0 monofilament suture is performed first for the proximal anastomosis (Figure 6). Subsequently, the distal anastomosis is performed after trimming the conduit to the appropriate length. Parachuting the distal end of the conduit and folding it over on itself to obtain some grade of tension (which will disappear after declamping) is advisable. When the blood flow is restored, the length of the conduit will increase. Attention must be taken to sidestep lengthening the PA, potentially causing kinking of the vessel, impaired blood flow, and, consequently, thrombus formation.

## 4. Technical Issues and Controversial Aspects of Perioperative Management

### 4.1. Surgical Incision

It is common to think that an extended surgical incision is preferable to perform complex bronchovascular reconstructions. Nevertheless, a minimally invasive approach including video-assisted thoracic surgery (VATS) is widely described for SL [36,37]. Due to the development of surgical skills and experience, posterolateral thoracotomy should be replaced by a muscle-sparing mini-thoracotomy. This surgical access allows the surgeon to perform a reconstructive procedure comfortably and safely with less pain and discomfort for patients. A VATS approach could consistently increase the operative time and potential risk rate for inexpert surgeons.

### 4.2. Steroids

Another aspect that still remains controversial is the use of postoperative steroids in patients undergoing bronchial reconstruction. The postoperative use of low-dose steroids after an SL with bronchial and vascular reconstruction has proven favorable in the authors’ experience because it may help to prevent anastomotic edema and secretion retention, thus reducing the risk of atelectasis and granuloma formation. Aerosolized steroids (methyl-prednisolone 5 mg twice a day) are also part of postoperative treatment. The short-term outcome of patients undergoing isolated PA reconstruction is generally not influenced by the use of steroids.

### 4.3. The Use of Viable Tissue Flaps

Bronchoarterial fistula is one of the most dangerous complications after lung reconstructive surgery, and a viable tissue flap, interposed between the bronchus and the PA, can effectively prevent the fistula occurrence. Researchers have described [38,39] the use of multiple options also including mediastinal fat pad, pericardial or pleural flap. However, the authors prefer to use an intercostal muscle flap thanks to its excellent vascularization provided by the intercostal artery [25].

At the opening of the chest, the intercostal muscle flap is prepared, avoiding inserting the rib retractor until the flap is not entirely obtained in order to not crush the intercostal vessels and interrupt the vascularization. After the incision of the periosteum of the fifth rib, it is separated from the bone together with the underlying intercostal muscle. Attention has to be paid to preserve the muscular insertion to the periosteum and to avoid injuring the intercostal neurovascular bundle. The intercostal muscle is then incised at the level of the superior margin of the sixth rib, and the flap is divided at the anterior insertion, ligating the pedicle. After completing the bronchial and the vascular anastomoses, a right-angle clamp is inserted between the PA and the bronchus, and the edge of the flap is slid backward to encircle the bronchial anastomosis, thus separating the bronchus from the PA. The pleural side of the flap should be in contact with the bronchial anastomosis and then fixed to the bronchus by interrupted absorbable 4-0 sutures.

### 4.4. Technical Tips for Avoiding Anastomotic Tension and Kinking of the PA

Avoiding tension on the anastomosis is one of the most important principles in bronchial and vascular reconstructive procedures.

If the anastomosis shows any grade of tension, it is advisable to complete the posterior portion of the suture and subsequently to parachute the arterial stumps together while lifting the lower lobe. In case of bronchovascular resection with patch reconstruction and bronchoplasty (end-to-end anastomosis), the axis of the bronchus is shortened, while the length of the PA remains unaltered. The PA may tend to kink and fold over on itself. This risk is increased by the repositioning of the PA due to the re-expansion of the lower lobe. Thrombosis may occur in this setting because of the impairment of blood flow. Cutting away the redundant segment and performing an end-to-end anastomosis could solve the problem in these circumstances.

A tension-free suture can be obtained by dividing the pulmonary ligament or, more frequently on the right side, opening the pericardium around the pulmonary vein during an end-to-end anastomosis reconstruction.

### 4.5. Choice of Prosthetic Material

According to the authors’ experience, the choice of the prosthetic tissue used for vascular reconstruction is a crucial factor. Autologous options may offer a number of advantages. If considering conduit reconstruction, the pulmonary vein is one of the best choices because it offers a “ready-to-use” conduit with tissue characteristics similar to those of the PA. The autologous vein conduit is fresh and unpreserved, cost-free, and biocompatible, but it has a limited length. Conversely, bovine pericardial tissue is less cost-effective and less biocompatible, but it has no limit in the amount of tissue. Adequate sizing of the conduit length represents a key point to the final success of the procedure.

If considering patch reconstruction, a recent analysis by Ebert et al. [40] has evaluated the risk of reintervention in a pediatric population based on the type of patch material. This paper has provided useful information for value-based decision making in the choice of prosthetic tissue. This study has retrospectively evaluated 214 pulmonary artery patch reconstruction procedures performed in a single center from 2000 to 2014. The reintervention rate after patch reconstruction was the primary outcome. A total number of 180 patients were considered with a median follow-up of 3.7 years. Patch materials were: patch homograft (*n* = 92), bovine pericardium (*n* = 44), autologous pericardium (*n* = 41), and porcine intestinal submucosal patch (*n* = 37). Median age and weight at the time of surgery were 12.1 months and 8.5 kg. Reintervention occurred at 34 sites (15.9%). According to a Cox proportional hazards regression, preoperative renal failure (HR = 4.36, CI 1.87–10.16, *p* < 0.001) and weight at surgery (HR = 0.93, CI 0.89–0.98, *p* = 0.004) were associated with reintervention, while the patch material did not influence the reintervention rate (*p* = 0.197). Cost per unit patch ranged from $0 for untreated autologous pericardium to $6105 for homograft patch. No relationship between the patch material used and reintervention rate was found. This finding combined with the widely different costs of patches may help surgeons in the choice of tissue to use for the patch.

### 4.6. Radiologic Evaluation after PA Reconstruction

To assess the patency of the reconstructed PA in the postoperative period, magnetic resonance angiography (MRA) or computed tomography (CT) scan with contrast medium and three-dimensional (3D) volume rendering may provide effective imaging of the PA. The noninvasive perfusion lung imaging is unnecessary if the patient is asymptomatic. During the oncological follow-up period, a CT scan with contrast medium is the proper diagnostic tool to evaluate the patency of the PA and its distal branching as well as the eventual relapse of the disease.

### 4.7. Anticoagulation Management

Proper anticoagulation management is a fundamental aspect when PA reconstruction is required and has not been adequately clarified by the initial reports [41]. In the past authors’ experience, systemic anticoagulation was initiated during the operation (3000–5000 U heparin sodium) [27] and maintained by the subcutaneous injection of heparin (16.000 U/d) for 7 to 10 days. Over time, the authors have progressively reduced the anticoagulation dose that is now administered as follows: intravenous injection of 1500 U sodium heparin during the resection phase without reversal by protamine sulfate at the end of the procedure [21] and 6000 U/d low-molecular-weight heparin administered subcutaneously for 7 days after surgery. Increased risk of thrombosis can be related to repositioning of the reconstructed PA due to the lower lobe re-expansion, which may cause its kinking and folding over on itself. The low-molecular-weight heparin helps to reduce the risk of thrombosis. Nevertheless, after patch or anastomotic reconstruction, the occurrence of thrombosis is an infrequent event. Differently, higher rates of thrombosis have been described after tangential resection with direct suture repair [41].

### 4.8. Postoperative Bleeding Avoidance

During surgery, sometimes, a leakage from the suture line may be unseen, since the PA is a low-pressure vessel. Therefore, a postoperative bleeding may appear on the first or second postoperative day after a patch or anastomotic reconstruction. A blood loss of as much as 800 to 1000 mL daily may occur after 1 or 2 days of no drainage, which may last for 1 or 2 days and then stop spontaneously, independently from anticoagulation. After unclamping and vessel distension, sutures too wide apart may result. An eventual bleeding would not appear immediately because the PA is stretched downward by the atelectatic lower lobe, and modest juxtaposition of the tissue edges is enough to counteract the low PA pressure. Nevertheless, postoperatively when the re-expansion of the lower lobe uplifts the hilum, the suture line may be distorted by the rotation and kinking of the PA, reopening the bleeding site. Therefore, checking the suture line carefully and assessing the PA position after re-expansion of the residual lobe is mandatory, particularly when using autologous pericardial patches.

### 4.9. Induction Therapy

SL can be performed safely and successfully after neoadjuvant therapy, without an increased complication rate [23,42,43,44,45,46]. After induction therapy, residual tumor or desmoplastic reaction (scarring tissue, or fibrosis) may involve the bronchus and/or PA at various extent. Concern about a higher complication rate in patients undergoing PA reconstruction after induction therapy has been proved to be unnecessary. In the authors’ experience, bronchial and PA reconstructive procedures can be performed in this setting [21], showing that there is no significant increase in the mortality and morbidity rates of patients undergoing SL after induction therapy. Moreover, the long-term survival of patients who underwent SL is better than that of patients who underwent PN after induction therapy. A recent study considering 35 patients out of 110 (31.8%) who received neoadjuvant chemotherapy has confirmed that the intraoperative performance and postoperative outcomes of double sleeve lobectomy following neoadjuvant chemotherapy were similar to those after direct surgery [47].

## 5. Complications and Perioperative Management

Primarily, a meticulous surgical technique is the main factor allowing a good final outcome in every reconstructive procedure of the PA. The induction therapy (especially chemoradiation) may theoretically increase the risk for postoperative complications because the vascularization and trophism of the tissues may be affected.

A fistula may be the result of technical anastomotic complications after bronchial or vascular reconstruction.

Tissue damage and fibrotic alteration induced by the neoadjuvant therapy may cause compromised healing and then anastomotic dehiscence. If considering operations with associated PA reconstruction, the most dangerous potentially life-threatening complication is represented by the broncho-arterial fistula.

The authors, based on their long-term experience, consider the protection of the bronchial anastomosis by viable tissue an effective procedure to minimize the risk of anastomotic dehiscence. The use of a vital vascularized flap may allow preventing the occurrence of bronchoarterial fistula, a rare but life-threatening event, especially in patients undergoing combined bronchial and PA reconstruction. The risk of PA erosion and consequent fatal bleeding may be minimized by the intercostal muscle flap interposition between the bronchus and the PA. Induction therapy seems not to significantly impair the intercostal muscle, representing the most adequate tissue for wrapping the reconstructed bronchus. Although a progressive heterotopic calcification of the muscular flap possibly causing a severe bronchial stenosis has been described in some papers, the present authors have not reported such a complication in their experience [39].

Alternative protective tissues have been reported to reduce the risk of dehiscence, such as mediastinal fat pad, pericardial or pleural flap, omentum, or internal mammary artery pedicles [25,38,39].

The complication risk in isolated PA reconstructive procedures seems not to be affected by induction therapy. In particular, bleeding and thrombosis (the two complications potentially due to a technical pitfall) are considered not to be significantly influenced by the induction treatment. As previously described, bleeding can occur later, after lung re-expansion, which may induce a change in the PA axis. Chemotherapy and radiotherapy may induce a reduction in the lung parenchyma elasticity. Therefore, the re-expansion of the residual lobe after induction treatment may require longer time compared to standard cases. In these cases, late unforeseen blood loss with spontaneous resolution within 24 to 48 h may be observed.

The postoperative complications rate in early experience by the present authors [27] was 13.4%. Complications occurred in seven out of 52 operated patients, including pulmonary edema (*n* = 1), arrhythmia (*n* = 1), empyema (*n* = 2), PA thrombosis (*n* = 1), and late bronchial stenosis in two patients who underwent double sleeve lobectomy. In the report by Shrager [41] et al., there were no perioperative or postoperative deaths. Early postoperative complications included two major complications in two patients (6.1%) and 20 minor complications in 13 patients (39%). Lausberg in 2005 [14] reported no other complications except for respiratory infections. One patient died from pneumonia and after died from sepsis on postoperative day 13, accounting for an early mortality of 1.5%. In the 29 angioplasty patients from Nagayasu’s study [46], there were 12 patch plasties and 17 sleeve plasties. The 90-day postoperative morbidity rate was 23.8% (35 of 147 patients) overall, 22.9% (27 of 118 patients) in the bronchoplasty group, and 27.6% (eight of 29 patients) in the broncho-angioplasty group (*p* = 0.5953). The 90-day postoperative mortality rate was 8.2% (12/147) overall, 5.9% (7/118) in the bronchoplasty group, and 17.2% (5/29) in the broncho-angioplasty group (*p* = 0.047). Cerfolio published in 2007 a report including 42 PA reconstructions [24]. Transient atrial fibrillation was the most common postoperative complication (*n* = 6/14.3%). Alifano in 2009 [45] reported a case-series of 93 PA reconstructions. There were no intraoperative deaths. Postoperative mortality was 5.4% (*n* = 5). Three deaths occurred after right lobectomy (two with bronchoplasty), and two occurred after left lobectomy. Postoperative pneumonia and acute respiratory distress (with no infection identified) were responsible for three deaths. In two further patients, acute myocardial infarction followed by intractable cardiogenic shock was the cause of death.

Among the analyzed factors, the following were associated with postoperative death at univariate analysis: age (*p* = 0.043), cardiovascular disease (*p* = 0.007), previous neoplasm (*p* = 0.030), *p* stage (*p* = 0.002), smoking (*p* = 0.020), and FEV1 (*p* = 0.035). Multivariate analysis showed that only cardiovascular disease (*p* = 0.022) and FEV1 (*p* = 0.039) were independent predictors of postoperative mortality. No statistical difference was found with respect to associated bronchoplasty or the extent of resection (lobectomy or bilobectomy).Other complications (not responsible for death) occurred in 27 (29.0%) other patients and were in some instances associated: postoperative pneumonia (*n* = 12), supraventricular arrhythmias (*n* = 5), persistent air leak without evident bronchopleural fistula (*n* = 5), respiratory failure (*n* = 5), atelectasia without pneumonia (*n* = 3), acute pericarditis (*n* = 1), and delirium (*n* = 1). One of the largest studies focusing on the present authors’ PA reconstruction experience was published by Venuta et in 2009 [28] reporting an overall morbidity of 28.5%. The procedure-related major complications were one PA thrombosis requiring completion pneumonectomy (occlusion of left PA demonstrated by angiography on postoperative day 2) and one massive hemoptysis leading to death (postoperative day 28); the latter complication occurred in a patient undergoing combined bronchovascular reconstruction. The operative mortality was 0.95%. Barthet in 2013 [31] reported a series with 32 PA reconstructions: 20 were end-to-end anastomosis, 2 were pericardial patch reconstructions, and 10 were PA replacements. Grafts originated from multiorgan donors whose PA and thoracic aorta were harvested, assessed, cryopreserved, and packed, as previously described. Reconstruction was performed in 2 patients by autologous pericardium patch, in 20 by end-to-end anastomosis, and in 10 by PA replacement by cryopreserved arterial allograft. There were no hospital deaths. Five patients sustained morbidity, including four major and two minor complications. One patient required prompt completion pneumonectomy because of thrombosis of the allograft on the first postoperative day. Cryopreserved allograft patency was 90% (9/10).

Results of PA reconstruction with pulmonary vein conduit were reported by the present authors [30]. The postoperative morbidity rate was 29.1% (one chylothorax, three atrial fibrillation, one parenchymal atelectasis, one pericarditis and one bleeding requiring rethoracotomy). No complications related to the reconstructive procedure occurred. There were no postoperative deaths. Complete patency of the reconstructed PA was seen in all patients on the postoperative contrast CT scans performed every 6 months. In particular, no significant stenosis, late aneurysmal problem or calcification of the reconstructed vessel was observed. The first CT scan to assess PA patency was performed at discharge.

## 6. Short-Term and Long-Term Results

Stage-by-stage survival of patients undergoing PA reconstruction is comparable to that reported in patients who underwent standard major lung resections [48,49]. The impact of nodal status on survival is likewise equivalent if comparing SL with standard resection. In case of N1 or N2 disease, once the decision to perform a resection with intent to cure is taken, PA reconstruction can also be proposed as an adequate procedure in this setting. Moreover, no significant difference has been reported between an isolated PA reconstruction and PA reconstruction associated with bronchial sleeve in terms of postoperative mortality and morbidity [46]. Furthermore, in a study published by the present authors, combined broncho-vascular reconstructions have been reported to have better survival compared with pneumonectomy [50]. This finding suggests that even these complex lung-sparing operations can be pursued with intent to cure as long as a complete anatomic resection is achieved [51].

According to the literature data, PA reconstruction associated with lung resection for lung cancer is safe and effective (Appendix A).

Distant and local recurrences rates show no difference between SL and standard major resection, irrespective of neoadjuvant treatment. The postoperative morbidity and mortality risk after induction therapy is significantly increased when performing PN, especially right PN. PN after induction therapy has showed mortality rates ranging between 14% and 43% according to results from several series [43,44,48]. This finding suggests that multimodality treatment, which is mandatory to achieve a locoregional control for advanced stage tumors, should not preclude the choice for reconstructive procedures.

When PA reconstruction is compared to PN, irrespective of neoadjuvant therapy, SL is constantly reported as a valid alternative, with lower short-term mortality and morbidity, without affecting long-term oncologic results. Schiavon et al. [52] published in 2021 a propensity score weighting study comparing lobectomy associated to PA reconstruction with PN for NSCLC. PN was associated with a higher 30-day and 90-day mortality rate (*p* = 0.02 and *p* = 0.03, respectively) as well as with a higher incidence of major complications (*p* = 0.004). Long-term results showed comparable outcomes for PN vs. SL if considering 5-year disease-free survival (52.2% vs. 46%, *p* = 0.57) and overall 5-year survival (41.9% vs. 35.6%, *p* = 0.57).

Recently, Hattori [53] et al. have reported that the 3-year overall survival was significantly better after extended SL (bronchial and vascular) compared with that after PN (62.8% vs. 45.2%, *p* = 0.047).

In another recent study from Yang et al. [54], analyzing survival outcomes, a log-rank test revealed no significant difference in overall survival (*p* = 0.381) and recurrence-free survival (*p* = 0.619) between the two surgical procedures (bronchial sleeve lobectomy with pulmonary arterioplasty and PN), thus confirming that this complex procedure is safe and reliable for centrally located non-small cell lung cancer concurrently involving the PA and bronchus.

One of the strongest indicators influencing the decision to perform an SL rather than a PN is the postoperative quality of life. Even though SL after induction therapy has not been adequately analyzed in this setting, a number of studies indicate that lung parenchyma sparing is related with better postoperative quality of life, determining a superior cardiopulmonary reserve, less pulmonary edema, and decreased right ventricular dysfunction caused by a lower pulmonary vascular resistance [16,18]. Ferguson and Lehman [17] proved through a meta-analysis that the quality-adjusted years quoted were 4.37 after SL and 2.48 after PN. A retrospective study Melloul et al. [55] showed significantly higher postoperative FEV1 values in patients who underwent SL. Martin-Ucar et al. [56] reported in a prospective study that mean FEV1 loss after parenchyma-sparing operations was 170 mL (range, 0–500 mL) compared with 600 mL (range, 200–1400 mL) after PN, proving a significant functional benefit for patients undergoing SL. The present authors have reported better functional and oncological outcomes after SL compared with PN in 2016 [57]. Furthermore, a number of studies from other groups all over the world [58,59,60] have confirmed the improved survival, quality of life, pulmonary function, and the lower mortality rates of both bronchial and arterial SL compared with PN.

## 7. Conclusions

PA reconstruction associated with lobectomy is a safe and viable parenchymal sparing intervention to radically treat lung cancer, allowing better long-term survival, lower perioperative morbidity and mortality rates and functional benefits if compared with PN.

## Figures and Tables

**Figure 1 cancers-14-04782-f001:**
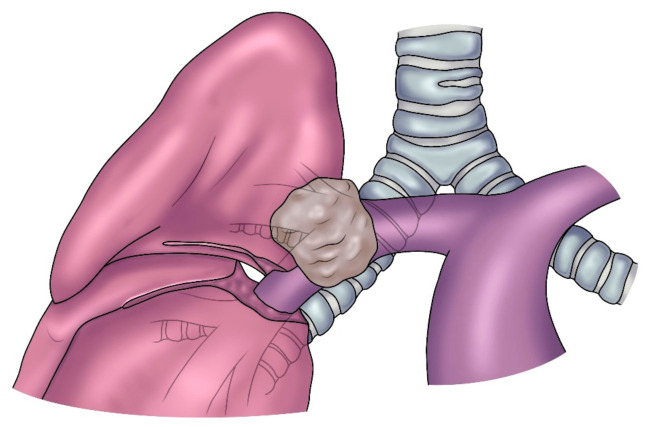
A tumor of the right upper lobe infiltrating the PA (posterior view).

**Figure 2 cancers-14-04782-f002:**
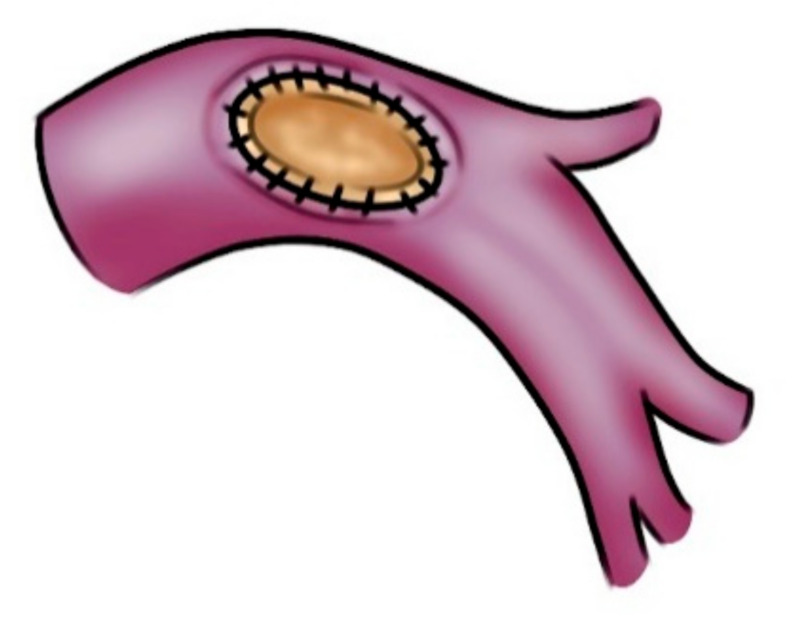
Left PA reconstruction with a patch of heterologous pericardium.

**Figure 3 cancers-14-04782-f003:**
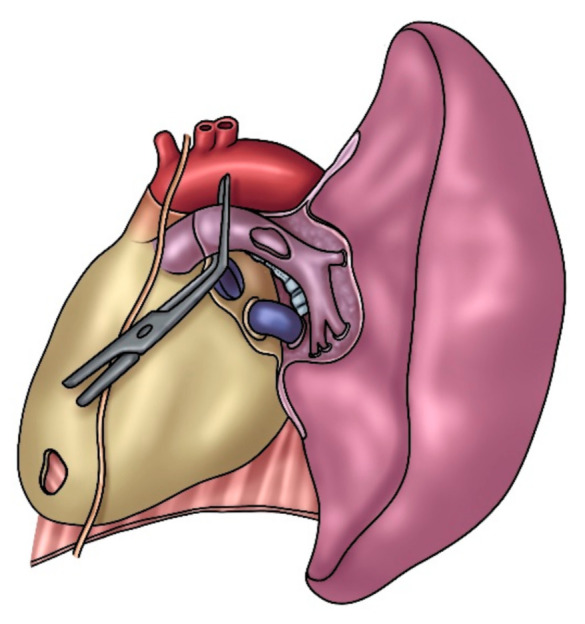
Left PA reconstruction with a patch of autologous pericardium. The pericardial defect is visible in the site of the leaflet harvesting.

**Figure 4 cancers-14-04782-f004:**
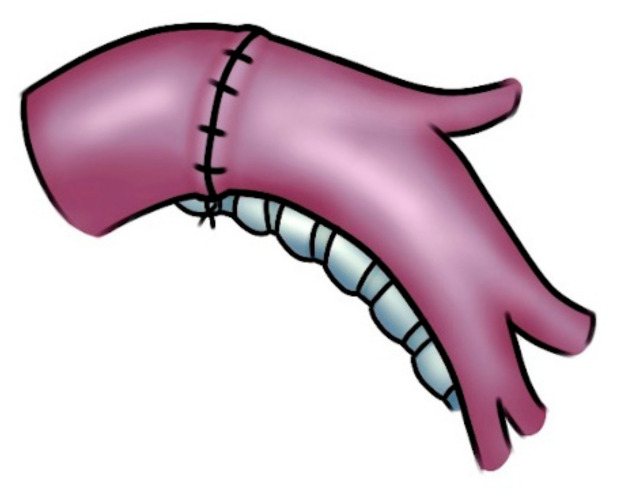
Left PA reconstruction with an end-to-end anastomosis.

**Figure 5 cancers-14-04782-f005:**
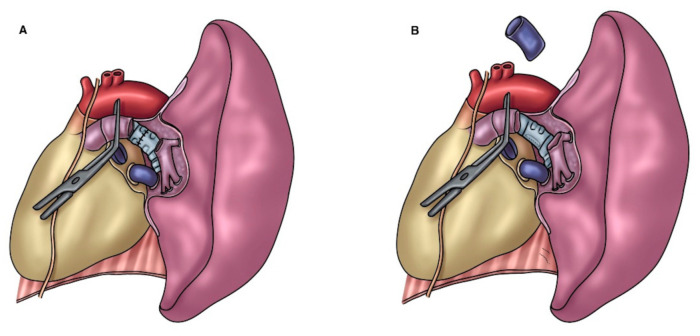
(**A**) Left PA defect ensued after a long-segment PA resection for a tumor of the left upper lobe; (**B**) Left PA reconstruction with an autologous pulmonary vein conduit.

**Figure 6 cancers-14-04782-f006:**
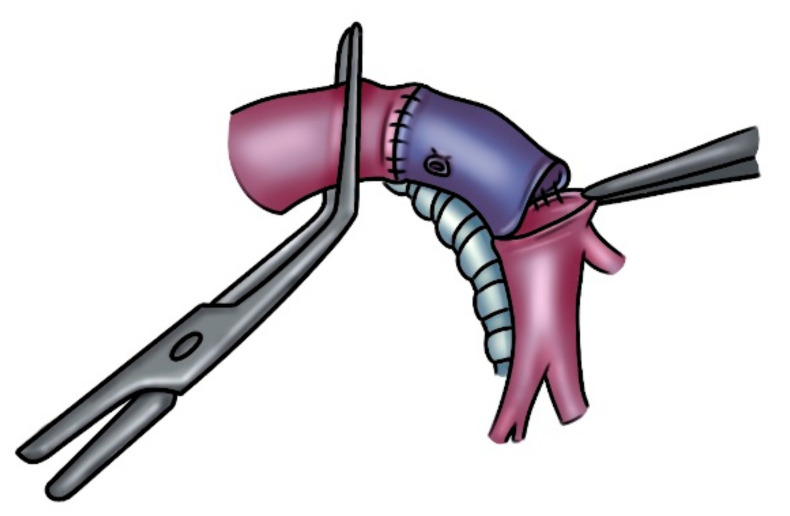
Left PA reconstruction with an autologous pulmonary vein conduit (proximal and distal anastomosis).

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
