# Peer review of "Parenchymal Sparing Surgery for Lung Cancer: Focus on Pulmonary Artery Reconstruction"

_cancers, 2022, doi:10.3390/cancers14194782_

Round 1

Reviewer 1 Report

PULMONARY ARTERY RESECTION AND RECONSTRUCTION FOR LUNG CANCER

An article which analyzed some of the main technical and oncological aspects regarding lung-sparing operations over time.

The overall structure of the manuscript is complete and the scientific question is clearly presented. The figures are clear and easy to understand.

Comments

• The data of patients undergoing a lung-sparing operation could be included to have better results

• It is not completely clear which studies are compared to each other

• In this case, Table 1 is not structured enough and confuses the reader

• It is also not completely clear if during the study period laboratory results were examined and discussed. In that case, it should be included, or better presented in the study.

• “bronchial sleeve resection” and “pulmonary artery reconstruction” in order to eliminate lung cancer in the title may underline better the purpose of the project

• A conclusion of the article is not drawn

Author Response

REVIEWER 1

1)….the authors must design several meaningful tables to summarize and compare the controversial issues.

Reply: a new table with results after PA reconstruction has been included. 

2)In the Indication and Preoperative Evaluation the authors did not summarize the indication and preoperative evaluation. It lost focus.

Reply: The “Indications and Preoperative Evaluation” paragraph has bee modified and improved with the addiction of several sentences in order to more extensively report and discuss the authors’ point of view and experience in this setting. 

3)The authors must review the technique and issue of bronchial reconstruction. The authors only focus on pulmonary artery reconstruction.

Reply: the following sentence has been included in the text (paragraph Sleeve Resection and Reconstruction by End- to-End Anastomosis):” Technical issues regarding bronchial reconstruction after SL have been extensively reported in previous publications by the present Authors [6,20,21,22] and are not repeated in this article due to its specific focus on PA reconstruction.”

4)The perioperative management is too short. The authors must analyze and summarize the complications and perioperative management by using more words.

Reply: Additional sentences reporting literature data regarding main complications have been included in the in the above mentioned paragraph.

Reviewer 2 Report

The authors reviewed technical and oncological aspects of sleeve lobectomy. The goal of the sleeve lobectomy is complete R0 resection. It comprised pulmonary artery and bronchial reconstruction. Several studies have demonstrated that sleeve lobectomy is superior to pneumonectomy. However, the authors must design several meaningful tables to summarize and compare the controversial issues. The authors only described only through textual narration.

In the Indications and Preoperative Evaluation, the authors did not summarize the indication and pre-operative evaluation. It lost focus.

The authors must review the technique and issue of bronchial reconstruction. The authors only focus on pulmonary artery reconstruction.

The peri-operative management is too short. The authors must analyze and summarize the complications and perioperative management by using more words.

Author Response

REVIEWER 2 

1) The data of patients undergoing a lung spring operation could be included to have better results

Reply: Additional data concerning results after parenchymal sparing operations with PA reconstruction have been included in the text and in a new table 

2)It is not completely clear which studies are compared to each other

Reply: comparisons of results reported between parenchymal sparing operations and pneumonectomy in literature studies have been explained and clarified. 

3)Table 1 is not structured enough and confuses the readers.

Reply: Additional table with results of parenchymal sparing operations with PA reconstruction has been provided. 

4)It is also not completely clear if during the study period laboratory results were examined and discussed. In that case, it should be included, or better presented in the study.

Reply: Since this is a review article, a study period has not been reported. 

5) “bronchial sleeve resection” and “pulmonary artery reconstruction” in order to eliminate lung cancer in the title may underline better the purpose of the project.

Reply: The present article specifically focuses on parenchymal sparing operations with PA reconstruction. In order to better underline this topic the title has been modified as follows:” PARENCHYMAL SPARING SURGERY FOR LUNG CANCER: FOCUS ON PULMONARY ARTERY RECONSTRUCTION”. 

6)A Conclusion of the article is not drawn.

Reply: A Conclusion has been included in the text.

Round 2

Reviewer 2 Report

I have re-considered the point-to-point reply according to my suggestion. It is appropriate to be accepted for publication.